# Inkjet Printing Magnetostrictive Materials for Structural Health Monitoring of Carbon Fibre-Reinforced Polymer Composite

**DOI:** 10.3390/s24144657

**Published:** 2024-07-18

**Authors:** Nisar Ahmed, Patrick J. Smith, Nicola A. Morley

**Affiliations:** 1Centre for Additive Manufacturing, Faculty of Engineering, University of Nottingham, Nottingham NG7 2RD, UK; 2Department of Materials Science and Engineering, University of Sheffield, Sheffield S1 3JD, UK; n.a.morley@sheffield.ac.uk; 3Department of Mechanical Engineering, University of Sheffield, Sheffield S1 3JD, UK; patrick.smith@sheffield.ac.uk

**Keywords:** magnetostriction, magnetite, nickel, ink dispersion, inkjet printing, JetLab 4

## Abstract

Inkjet printing of magnetic materials has increased in recent years, as it has the potential to improve research in smart, functional materials. Magnetostriction is an inherent property of magnetic materials which allows strain or magnetic fields to be detected. This makes it very attractive for sensors in the area of structural health monitoring by detecting internal strains in carbon fibre-reinforced polymer (CFRP) composite. Inkjet printing offers design flexibility for these sensors to influence the magnetic response to the strain. This allows the sensor to be tailored to suit the location of defects in the CFRP. This research has looked into the viability of printable soft magnetic materials for structural health monitoring (SHM) of CFRP. Magnetite and nickel ink dispersions were selected to print using the JetLab 4 drop-on-demand technique. The printability of both inks was tested by selecting substrate, viscosity and solvent evaporation. Clogging was found to be an issue for both ink dispersions. Sonicating and adjusting the jetting parameters helped in distributing the nanoparticles. We found that magnetite nanoparticles were ideal as a sensor as there is more than double increase in saturation magnetisation by 49 Am^2^/kg and more than quadruple reduction of coercive field of 5.34 kA/m than nickel. The coil design was found to be the most sensitive to the field as a function of strain, where the gradient was around 80% higher than other sensor designs. Additive layering of 10, 20 and 30 layers of a magnetite square patch was investigated, and it was found that the 20-layered magnetite print had an improved field response to strain while maintaining excellent print resolution. SHM of CFRP was performed by inducing a strain via bending and it was found that the magnetite coil detected a change in field as the strain was applied.

## 1. Introduction

Printed electronics via inkjet printing onto substrates such as on printed circuit boards (PCB) have been around for decades. This method has gained an interest in printing smart magnetic materials for sensors and actuators [1]. One area of research is printing magnetostrictive sensors for structural health monitoring (SHM) of carbon fibre-reinforced polymer (CFRP) composite. The current state of the art in SHM shown in [2,3,4,5] conclude that fibre Bragg grating (FBG) and acoustic methods are suitable for SHM due to reliability in sensing damages in CFRP, whereas magnetostrictive sensors are limited in manufacturing methods and cost.

Therefore, this paper exploits the gap in reducing the manufacturing process time and cost of materials used for SHM devices via inkjet printing soft magnetic materials. Printing magnetostrictive materials has been difficult due to the materials’ availability and compatibility, which so far has been restricted to three-dimensional (3D) extrusion or powdered metal additive manufacturing [6,7]. Inkjet printing is desirable in applications where the weight and cost of material are important. The objective of this paper is to demonstrate the printability of magnetostrictive sensors and be able to monitor damages in aircraft CFRP material. The aim is to develop and print magnetostrictive via an inkjet printing technique; this allows additional freedom in designing various patterns and layers, which could improve the performance of the sensor. Therefore, this paper will print, characterise and develop a tension to test the sensor’s magnetic field response to strain.

In the literature, ref. [8] have shown that magnetostrictive materials can detect damages present in CFRP under bending and impact forces. One of the main advantages of using magnetostrictive materials is that magnetostriction is an inherent property of magnetic materials and does not degrade the CFRP. Ref. [9] have designed and developed an SHM device consisting of a magnetostrictive ribbon embedded in the CFRP while detecting the magnetic field using a coil inductor. Wireless monitoring can also be implemented and will be useful in aerospace applications. Another method shown by [10], made a printable copper inductor coil, which was used for detecting small changes in the magnetic field from the magnetostrictive sensor.

The two main forms of magnetoelastic effects [11] can be described by Equation (1): Joule magnetostriction, where the material changes in length under a magnetic field, and Equation (2): Inverse-Joule or Villari Effect, where the magnetic flux density changes direction under an applied strain.
(1)$$\varepsilon =\sigma ∕E_{y}^{H}+d_{33}H$$
(2)$$B=d_{33}\sigma +\mu^{\sigma }H$$
where in Equation (1), *ε* is the strain, *σ* is stress, $E_{y}^{H}$ is the compliance coefficient at constant field strength, *d*_33_ = *dε*/*dH* (strain/field), and *H* is the magnetic field. Where in Equation (2), *B* is the magnetic flux density in Tesla (T), *d*_33_ = *dB*/*dσ* (induction/stress), which is the magnetostrictive constant, and *μσ* is the permeability at constant mechanical stress.

Fe-based soft magnetic materials are ideal for structural health monitoring [6] as they have low coercivity and high saturation magnetisation. However, inkjet-printed magnetostrictive materials have not yet been tested for SHM of CFRP composite.

Printing magnetic materials by inkjet printing is achievable; however, there are certain factors that could influence the base metal [12,13]:Nozzle and droplets;Alignment of moments;Magnetic and non-magnetic printing.

Magnetic inks can form long liquid bridges from the nozzle, which makes the drop on demand (DOD) system difficult to print with magnetic inks. The inkjet system uses a piezoelectric actuator to push inks from the ink chamber to the nozzle therefore controlling the number of drops to the substrate. However, ref. [12], mentioned that magnetic inks could be pushed by a magnetic field instead of an actuator or any mechanical device. This would mean controlling the magnetic particles further by ensuring each drop contains magnetic metal particles and breaking the bridge with the drop to the nozzle, which could improve densification and further control of the droplets.

Thick film magnetostrictive material has been produced and measured by [14]. They have produced a paste containing Terfenol-D and glass frit, then printed using screen printing. By measuring the magnetostriction using the change in length from lasers, they reported that it has a lower magnetostriction constant than the bulk materials due to void formation in the thick film.

Nickel and magnetite are soft magnetic materials (ferromagnetic) that can be used for printing magnetostrictive sensors. In literature, nickel and magnetite nanoparticles have been used and successfully printed/deposited by [12,15,16,17,18,19,20,21,22,23,24]. Although nickel and magnetite have been shown to be printable, there have been issues with ink synthesis, droplet formation, substrate and post-printing.

There are various ways to synthesise magnetite and nickel inks, such as additives, sol–gel and solvents to improve the printing and post-printing process. The substrate onto which the ink is printed plays a key role in retaining the nanoparticles and evaporating the solvent. Ref. [20], have shown that nickel can be printed onto glass slides using a Dimatix Materials Printer inkjet system for electronic applications. The nickel inks printed were preheated to 140 °C on a hotplate and then heated to 180 °C after printing to avoid wetting and to sufficiently evaporate the solvent. However, glass as a substrate material cannot be used as a magnetostrictive sensor as there needs to be greater flexibility in the substrate. Polymer materials can be used to print nickel ink; however, there are limitations; for example, ref. [24], shows that the PET substrate material causes shrinkage and blistering problems when curing nickel inks. Kapton film could be used as the melting point for Kapton is around 400 °C, well above the curing temperature of nickel. Ref. [19], have shown that nickel ink (with ethanol solvent) can be printed on Kapton and cured on a hot plate at 150 °C for 20 min and then flashlight sintered.

Ref. [23] have printed magnetite inks via inkjet printing onto sacrificial paper to produce a turntable actuator and explored the effect of synthesising magnetite solvent to form a photo-curable ink such as oleic acid. Polymeric resin is used to strengthen and form bonds between magnetite nanoparticles. The magnetic properties at room temperature were measured for both magnetite inks in the original and with polymeric resin. It was shown that the polymeric resin reduced the magnetic properties, such as saturation magnetisation (M_s_), slightly, which was due to the higher level of non-magnetic material within the print. Ref. [22] explored the effect of printing magnetite inks onto paper by changing the jetting (piezoelectric or thermal), temperature of print and size of magnetite nanoparticles. The magnetic properties remain unchanged for both piezoelectric and thermal print heads. Ref. [18] produced an inkjet-printed magnetite inductor core on paper at room temperature and polyimide (sintered at 300 °C) by JetLab 4 by MicroFab Inc. (Plano, TX, USA). Oleic acid was used to cover the ink and was treated with potassium hydroxide. Aggregation in some areas was formed; however, the print was successful on both paper and polyimide.

## 2. Methods

The metal dispersion inks that were selected to be studied in this research were 20% Magnetite with Dimethylformamide (DMF) and 2% Nickel with N-Methyl-2-pyrrolidone (NMP) nanoparticles (NPs) inks, manufactured by Nanoshel. Table 1 shows the composition and physical properties of the inks. These include the solvent within the ink, the viscosity of the ink, surface tension, particle size and solvent evaporation temperature. The composition and physical properties are vital for printing NPs using an inkjet system to avoid clogging and printability on a substrate. For example, particle size above 100 nm would be prone to clogging depending on the nozzle size. The ink composition and particles were readily available and tested by the manufacturer. However, for inkjet printing applications, it is critical to know the viscosity of the ink and tailor the composition for the printing system. Equation (3) shows the Z number that determines the printability of the inks [25,26].
(3)$$Z=\frac{\sqrt{\rho d_{o}\gamma_{t}}}{\mu_{v}}$$
where *ρ* is density, $d_{o}$ is the orifice diameter, *γ_t_* is the surface tension of the ink, and *µ_v_* is the viscosity of the ink. An ink can be considered printable with a *Z* number to be between 1 to 10.

A viscometer was used to measure the viscosity of both inks. The viscometer was calibrated by a one-point calibration technique with water at room temperature before measuring the inks. Distilled water was measured at a known temperature and then compared to published values, as shown in Table A1 [27]. Magnetite and nickel ink were placed in a polycarbonate container and placed in a sample holder. The viscosity was measured by placing two vibrating probes and a thermocouple lowered and aligned to the meniscus of the ink.

### 2.1. Printing Process

The experimental procedure for printing the metal NPs in the JetLab 4 was to first prepare the ink. An ultrasonic bath was used for 30 min before transferring approximately 3 mL of ink to the ink reservoir. The reservoir was then placed into the JetLab 4 printing machine. The jetting parameters were then calibrated. Before printing onto the substrates, they were cleaned by air drying to remove any dust or impurities.

The piezoelectric print head was bought from MicroFab Inc. (Plano, TX, USA); it had a nozzle size of 60 µm with controllable voltage output. The printing parameters used were a standard wave at a rise time of 5 µs, dwell time of 5 µs, fall time of 50 µs, echo time of 6 µs, rise time of 10 µs, idle voltage of 0 V, dwell voltage of 60 V and echo voltage of −60 V as seen in Figure 1. There is a slight variation in these parameters depending on the ink mixture. For example, the dwell voltage may change to 65 V, or the dwell time may change to 3 µs to prevent clogging of satellites in each droplet. For example, nickel ink has heavier metal particles, which were not easily mixed into the solution, therefore creating an additional issue when there was a difference in the density at the nozzle. This created a clogging problem where the jetting parameters were not calibrated for the heavier particles of the solution. This results in an irregular print, which cannot be altered during printing.

Designs for printing were made in Ansys CAD Space Claim 2019 R2 and converted into bitmap monochrome file. The resolution of the image depends on the number of pixels in the image.

### 2.2. Post Treatment

After the printing process, the printed design required further treatment in order to be used as a sensor. The first stage involved drying the print after the printing process, which involved heating to evaporate the solvent. The print was left at room temperature, leaving the metal NPs on the substrate.

Coating is another treatment, which serves to protect the inkjet-printed designs from external environments. Two ways of applying a coating were used, these were by applying a layer of silicone via spin coater or spray coating a layer of acrylic polymer. Both coating methods have advantages and disadvantages of applying and protecting the print. Spray coating is an easy way of applying and protecting the print, as it needs no additional machines or equipment. However, the spin coater ensures an even coating, which is useful without compromising thickness and detection of the magnetic field.

An Ossila spin coating machine was used to apply a layer of polydimethylsiloxane (PDMS) to the print under a fume hood. The PDMS (Slygard 184) used was purchased from Merck Life Science (Dorset, UK), which came in pre-packed solutions of resin and hardener. As the PDMS solution was viscous, the spin coater was set at 3000 RPM for 12 s. The coated print was left to dry at room temperature for 24 h. The print with the PDMS layer was then placed in an oven at 60 °C for 1 h to cure. An acrylic conformal coating spray was purchased from RS component. The spray was used under a fume hood and sprayed directly onto the print from 15 cm away.

### 2.3. Production of Carbon Fibre Composite Sample

The carbon fibre prepreg 4-ply twill weave (VTC401-C200T-HS-3K-42%RW) was used in this project and was supplied by SHD Composites Ltd. (Lincolnshire, UK). The composite laminate was formed using vacuum bagging. The prepreg CFRP were cut to size and layered on top of each other on a glass substrate and sealed in a vacuum bag at −28 Hg. The autoclave was used to apply pressure of 6 bar and heated to 60 °C at a rate of 3 °C/min and held for 60 min. Then the temperature increased to 120 °C at a rate of 3 °C/min for 60 min and then cooled down to room temperature. The cure cycle was recommended for the specific carbon fibre prepreg (VTC-401) from the supplier [28], as demonstrated in [9]. A tile cutter was used to cut the CFRP to size. This included a blade submerged in water to prevent excess dust from cutting the CFRP samples. For the strain bending test, the laminates were made to a size of 25 mm × 50 mm × 0.75 mm

### 2.4. Villari Effect Magnetostriction

#### Strain Bending Test

An inductance measurement was used to measure the magnetoelastic performance of the soft magnetic printed designs under strain. These methods were selected from literature as they have proven to detect magnetic fields. The magnetic prints were strained by placing them on a bending rig to apply a bending force. The bending test setup shown in Figure 2a shows the force applied on two ends, which forces the sample around a known curvature radius. The dimensions of the bend rig and radius of curvature were measured and produced in the Ansys CAD design package. The CAD design was converted to an STL file for printing. A resin photo-polymer printer was used to print 3D bend rigs, as shown in Figure 2b. As the print is at the top of the paper, the bend will produce a tensile force and reorientate the magnetic moments, therefore producing a change in the magnetic dipole and field.

The radius of curvature is converted to strain by using Equation (4). Where *ε* is strain, *y* is the distance from the neutral axis, and *R* is the radius of curvature. By converting the radius of curvature of R1000, R900, R800, R700, R600, R500, R400, R300, R200 and R100 (where R1000 = 1000 mm radius) to strain values of 0.13, 0.14, 0.16, 0.19, 0.22, 0.26, 0.33, 0.43, 0.65 and 1.3 µε respectively for paper substrate. The strain depends on the distance of curvature from the neutral axis; therefore, calculated values for paper and CFRP are presented in Appendix A in Figure A1. The 3D-printed bending rig dimensions were measured again after printing for recalculation of the strain.
(4)$$\varepsilon =\frac{y}{R}$$
(5)L=ΦI
(6)$$L=\mu_{0}\;\frac{N^{2}A}{l}$$

Measuring a change in inductance is a direct way of measuring the change in the magnetic field of a material. For this measurement, a coil was made to measure the change in inductance when the strain was applied to the print. The inductance is proportional to magnetic flux, as shown in Equation (5), where L is in Henries, Φ is magnetic flux, and I is the current. Therefore, the magnetic flux can be derived from the inductance at a constant current in the coil [29]. This allows flexible measurement to adjust the sensitivity of the inductance. In Equation (6), where L is in Henries, μ_0_ is the permeability of free space, *N* is the number of turns, *A* is the inner core area in m^2^, l is the length of the coil in metres. This shows that in a coil, the number of turns increases the inductance greatly, therefore increasing the area. The dimensions of the 3D-printed coil holder are shown in Figure 3a, where the diameter of the inner air core is 5 mm, and the diameter of the coil holder is 16 mm. The inductor was made from copper wire of 0.1 mm thickness and an air core size radius of 1.5 mm.

The experimental setup to measure the inductance of the printed sample consisted of an 891-bench top LCR meter from BK precision, connected to a coil inductor, as seen in Figure 3b,c. A current was applied at a frequency of 1 kHz with a voltage of 1 vrms for the benchtop LCR meter.

## 3. Results and Discussion

### 3.1. Magnetic Properties

The magnetic properties were measured by measuring the hysteresis loop using a Quantum Design MPMS magnetometer at room temperature, as shown in Figure 4 and Table 2. From the hysteresis loops, it was observed that the magnetite NP had more than double the saturation magnetisation (M_s_) and a lower coercivity (H_c_) and remanence (M_r_) than the nickel NP, which is preferable for sensing application.

From the literature, the magnetic properties of both magnetite and nickel NP have been measured [15,16]. The M_s_ of the magnetite was 76 Am^2^/kg, which is close to published values at 70–80 Am^2^/kg. There are similarities in magnetic properties for all literature shown except for one published paper [15], where H_c_ and M_r_ show a 10 kA/m and 10 Am^2^/kg difference, respectively, from the current work and other published papers. This may be due to the technique used by [15]; they used an electrode bath (electrolysis) to form magnetite, which may have reduced the particle size, hence increasing the H_c_ and M_r_ values considerably. It is reported that bulk magnetite has M_s_ at 92 Am^2^/kg [15], which is similar to the measured magnetite M_s_ in Table 2, as the difference is due to surface effects such as spin canting.

Whereas the published values and current work for Nickel NPs [30,31] show that there is considerable variation in the magnetic properties. For example, in [30], the data is closest to this work where the M_s_ is around 30 Am^2^/kg; however, in [31], the M_s_ is around 55 Am^2^/kg, which was higher than all published and current work. The gap of 20 Am^2^/kg is significant, more than the published values for magnetite. The size and composition of NPs of nickel do affect the M_s_ value, which could explain the high variation between the published values. For example, a particle size of 20 nm has a single domain, whereas above this size, they are multi-domain. For example, ref. [31], reported that the hysteresis loop for Nickel NPs with a diameter of 22 nm has a similar M_s_ value of around 30 Am^2^/kg from the hysteresis loop. However, this work has a particle diameter of 80 nm, far larger than in [31]. This may be down to their process in obtaining Nickel NPs, where thermal fluctuation, solvent and technique could affect their magnetic properties.

### 3.2. Viscosity Measurement

Table 3 shows the result of magnetite and nickel ink viscosity at room temperature measured using a viscometer SV–1 A from A&D Company Ltd. (Tokyo, Japan). This shows that both inks are suitable for inkjet printing as the value is below 20 cP and above 1 cP. However, the measurement was taken at the meniscus, where there is more solvent than heavy NPs. As the heavy NPs tend to drop to the bottom of the polycarbonate container. This creates an issue when measuring, as the measured value is not the true viscosity of the ink; rather, it is more of the viscosity of the solvent. Therefore, the process of setting up the measurement was performed quickly to prevent the heavy NPs from dropping to the bottom before the viscosity measurement. The true viscosity is likely to be higher. The viscosity for nickel is slightly higher than magnetite ink even though magnetite contains 20 wt.% NPs and Nickel has 2 wt.% NPs in the ink. The higher viscosity is due to the higher molecular weight and density of the solvent, as NMP in nickel ink has a higher density than DMF in magnetite ink.

The Z number was calculated by using the equation in [26] and found to be around 6–8 cP for nickel ink and 9.7–12.9 cP for magnetite ink. However, the inks can be diluted to improve the viscosity and reduce the Z number. The Z number is less or around the critical value of 10 cP; therefore, this makes it suitable for inkjet printing. In the published paper by [32], the magnetite dispersion from the same company was shown to have a viscosity value of around 1.6–1.7 cP at 297 K. Although the test was not performed at around 297 K, the viscosity in the published values would be slightly lower than in this work. Nevertheless, the inks have shown a good level of viscosity for inkjet printing using the JetLab 4 printing system.

### 3.3. Print Analysis

Figure 5 shows the (a) image and (b) printed design for (i) magnetite uniaxial patch, (ii) magnetite coil, (iii) magnetite 3 mm grid design and (iiiv) nickel lines on photo paper printed by JetLab 4 with 60 µm print head orifice, respectively. All designs are measured at 25 × 25 mm^2^ in length. The grid designs are adjusted by changing the infill size as track gaps are increased from 3 mm to 5 mm in distance.

The microscope image of magnetite print on photo paper presented in Figure 6a shows that each droplet has a non-spherical shape, but there is good adhesion between the magnetite nanoparticles on paper. Some droplets, as seen in Figure 6a, have joined, which may have been due to the droplets being too close together during the printing process; this could be improved by adjusting the jetting parameters to reduce satellite droplets, as seen in Figure A2. The radius of droplets (R1, R2 and R3) are all similar to each other with a mean of 78.20 ± 2.5 µm, while the diameter (D1, D2 and D3) have a mean of 159.45 ± 5.3 µm, which is due to the misalignment of droplets. Although the droplets are shown to have a slight variation in diameter and radius, the size of the droplets is relatively good. This would not affect the magnetisation change when strain is applied.

The nickel coil is printed as seen in Figure 6b, which shows the microscope image of nickel ink on photo paper. The microscope image shows that the nickel drops are erratic and do not show good cohesion on the paper. Compared to magnetite, where multiple prints show a relatively good accuracy (print direction on top of each drop), nickel does not print well in additive layering as each drop does not overlap each other. There are areas where the print is inconsistent, as clogging may have disrupted the printing process. The size of the droplets shown in Figure 6b has a mean diameter of 141.6 ± 2.3 µm (D1, D2 and D3) and a spherical mean radius of 70.7 ± 1.5 µm (R1, R2 and R3). In comparison, the nickel drop measured shows that the drops are about 20 µm smaller than magnetite droplets, which is due to the lower concentration of nickel in each droplet and are prone to forming voids and non-uniform NPs due to excess solvent in each drop.

The magnetite print has a resolution of around 132 dots per inch (DPI), whereas in the literature [11], the resolution is much higher at 185 DPI. Reducing the droplet size can be achieved by reducing the wettability of the ink on the substrate. Layer-by-layer printing depends on the evaporation of the solvent to prevent the NPs from spreading. Therefore, applying heat during printing which increases solvent evaporation and reduces migration of NPs to the edge increases the density of the NPs in the droplet.

#### SEM and EDS Analysis

Magnetite and nickel NPs were further analysed by SEM and EDS using AZtec one software (https://nano.oxinst.com/products/aztec/aztecone (accessed on 30 September 2023)). Figure 7 shows the SEM and EDS analysis of magnetite print on paper where (a) and (c) show the magnetite droplet on paper substrate. The distribution of magnetite NPs is not homogeneous across the droplet, as larger amalgamated NPs can be seen clearly in the SEM image. This will affect the roughness and height of the droplet. Achieving homogeneous droplets is essential to reduce porosity and controllable anisotropy. Instead, the droplet will have random alignment if the porosity increases. The EDS image in (b) and spectrum in (d) and (e) shows elements of iron, carbon and oxygen present in the droplet. Iron elements in magnetite droplets can be clearly seen. Carbon and oxygen can be seen all over the substrate and the droplet, as oxygen is present in magnetite and paper, while carbon is present in paper but not in magnetite; however, due to the thinness of the magnetite drops, the electron beam is likely also to be detecting the substrate as well as the droplets.

Nickel droplets on paper SEM and EDS mapping/spectrum can be seen in Figure 8. Compared to magnetite, nickel NPs on paper contain higher levels of porosity, which can be clearly seen in the SEM layered image in (a). This is due to the higher level of solvent than magnetite which is prone to porosity when it is evaporated or absorbed into the paper. The EDS mapping in (b) shows elements of nickel, oxygen, silicon and carbon present. The droplet shows a clear presence of nickel and carbon elements, whereas silicon and oxygen are present in the paper. This is as expected as the nickel NPs contain only nickel and paper, showing oxygen. Spectrum (c) and (d) show that the SEM image in (a) contained a majority of nickel and oxygen.

### 3.4. Inkjet Printing Sensor Measurement

The graph in Figure 9a shows the mean value of magnetite print inductance, measured on each bend rig, while Figure 9b shows the inductance as a function of strain for nickel print with a deviation of ±0.005 µH. Most of the results shown in Figure 9 appear to show a negative inductance as the strain is increased. As the strain is increased, the distance between the coil and magnetite print is increased and the tension caused by the bend rig would align the moments toward the stress direction, which would naturally assume a negative trend. The measured magnetic field may be improved by increasing the density or layers of the print. In comparison, nickel has shown a higher inductance than magnetite; for example, the coil design in nickel at 0.22 µε is at 0.025 ΔµH compared to 0.005 ΔµH for magnetite, as shown in Figure 9a,b.

A linear fit was performed on the data where Figure 10 shows the calculated linear fitting of the magnetite print for the graph (a) intercept, (b) slope and (c) R-squared values. The gradient is the change in the magnetic field as a function of strain applied; therefore, the greater the change, the better the sensor design performs as a function of strain. The intercept shows the initial inductance value where no strain is applied, which may be significant to compare with other designs such as increased layers. The R-squared value shows how well the data fits the linear fitting and if increasing strain changes the magnetic field; for example, if the data is erratic, then the R-squared value is low. For a reliable sensor, an R-squared value should be close to 1.

The magnetite design that is most sensitive to strain is the coil design, as shown in Figure 10b. In comparison with other designs tested, the gradient of the coil design is lower, which may be due to the design itself. The R square value (COD), as seen in (c), shows that 80% of the data is due to the applied strain. In comparison, the R-value for a 5 mm grid, a 3 mm grid and a uniaxial patch are close to 0, which suggests that the change in strain does not affect the change in inductance. Data could be improved by gradually reducing the radius of curvature while measuring the inductance. For example, the intercept shown in (a) should show a positive value due to the demagnetising field of the printed NPs. The negative intercept could be due to the initial applied strain; in other words, the starting strain could be too high, resulting in a negative reduction in inductance.

To improve the change in the field with strain, different designs, the number of layers and print directions were explored, as seen in Figure 10b. The effect of changing the design from 5 mm to 3 mm and the uniaxial patch does not show a significant change. There is a slight increase in gradient and R squared value for the 5 mm grid. However, there is a larger deviation in the 3 mm grid than the 5 mm grid design, which may benefit from additional data points; the same can be said with the uniaxial patch. In comparison with the desktop printing grid design, the change in field to strain (cantilever) did show a difference between the 5 mm and 3 mm track gap design, as seen in [6]. Printing multiple layers does make a difference to the magnetoelastic performance. A coil with 10 layers shows an increased slope in the field as a function of strain compared to the single-layered coil, which is expected due to a greater amount of material deposited on the substrate. Even though the gradient for both single and multiple coil layers shows a negative tread, the addition of multiple layering causes an increased gradient to the magnetoelastic performance.

The print that is perpendicular to the strain direction shows the greatest sensitivity to inductance as a function of strain, as seen in Figure 10. The 100 × 200 px rectangular print (perpendicular to strain) has a higher inductance to strain sensitivity than the 200 × 100 px rectangle (parallel to strain) print. The intercept in (a) is shown to be almost the same for both print in parallel and perpendicular lines. As strain is applied, the perpendicular print produces a positive gradient, whereas the parallel print remains at zero, as seen in (b). It is true that the perpendicular gives a higher gradient but at lower to higher strain values, e.g., from 0.65 µε to 1.3 µε. The parallel print is more sensitive to the field as a function of strain at a lower strain value than the perpendicular direction. This may be due to the shape anisotropy effect where the alignment of print direction in the parallel, the easy direction is along the axis, which is why the inductance peaked much earlier than the perpendicular direction. The perpendicular direction peaked at 1.3 µε, which is due to the hard direction (as the alignment is already aligned towards the perpendicular direction). In Figure 6a, the microscope image of the printed magnetite on photo paper shows that individual drops of the magnetite ink are near-spherical shape, which suggests that the shape anisotropy is down to the direction of print and not due to the shape of individual droplets.

The graphs in Figure 11 show the fitting for nickel with parameters of (a) intercept, (b) gradient and (c) R squared value. Figure 11a shows that even though the intercept is higher, all the designs have smaller gradients in comparison to magnetite print, as shown in Figure 10b. For example, the nickel 5 mm grid and coil designs gave a good response to inductance as a function of strain. However, they have a lower gradient than magnetite coil and a 5 mm grid design. Although the gradient is very small for all designs, it was expected that the coil design would be more sensitive in nickel ink based on the design change rather than the material change. The change in magnetic field can change depending on the design and applied strain, which could be useful in various applications. For example, from Figure 9b, the coil design shows a much steadier and more gradual slope than the 5 mm grid, similar to a magnetite coil as strain is increased. For example, in the 5 mm grid design, from 0.22 µε to 0.33 µε, there is a small positive change in inductance, but when the strain is increased to 0.65 µε, there is a steep reduction in inductance. In comparison, the coil design shows a reduction of inductance from 0.22 µε to 0.65 µε. Therefore, the coil design has a better range of sensing for small and larger strains. The R-square values in (c) in both Figure 10 and Figure 11 show that for both magnetite and nickel, the uniaxial patch is the least effective design, as the data is not influenced by the strain size. This could be due to the print direction (perpendicular to strain) and gaps within the design, which could reduce the change in the field.

Overall, the design that gives the best magnetoelastic response is the coil design. While it does show a negative inductance gradient in both nickel and magnetite material, both do show a good trend of change in inductance to strain. The trend shows a gradual change, whereas other designs give a sharp and unpredictable change, which can distort the results. Therefore, this makes it ideal to select it as a sensor in future work. The rate of change in field as a function of strain is higher in magnetite than nickel, even though nickel has a higher intercept (demagnetising field) than magnetite print. Therefore, the material of choice in this work is magnetite.

#### 3.4.1. Additive Layering

Printing magnetite in layers has been shown to improve the NPs change in the field as a function of strain. However, this was performed with one design, comparing the single and 10-layer printed magnetite coil on photo paper. Printing a simple design, such as a square shape (5 × 5 mm^2^), makes it easier to print in layers of 10, 20 and 30.

Figure 12 shows the magnetite square design (5 × 5 mm^2^) in (a) 30, (b) 20 and (c) 10 layers. Significant differences can be seen in each layering print. For example, the 10 layers show clear print direction across the square design, whereas the 30 layers show erratic print direction, thus reducing the printing accuracy. This may be due to the excess satellites and clogging forming during printing, where the excess droplets can be seen on the edges of the square design. Therefore, when printing beyond 30 layers, the jetting parameters may need to be recalibrated in order to prevent satellites or excess droplets. The 20-layered print has been shown to have good definition and density to form an accurate square patch on the photo paper substrate.

Figure 13 shows the bending test performed for each layered magnetite sample in Figure 12. The radius of curvature was reduced from 600 mm to 1000 mm; this was used to see the effect of applying a smaller strain to the magnetite designs and whether there was any change in inductance. Not all square prints show a change in inductance or reliability as a function of strain. From all three sample square prints, on average, the change in inductance increased from an applied strain of 0.13 µε to 0.2 µε, and then the change in inductance reduced as the applied strain increased from 0.25 µε. This pattern was present in all print layers. However, the error seen in 10 and 30 layers exceeds the change in inductance, whereas in 20-layered square patches, the error is lower than the change in inductance to strain. All three prints have shown different levels of noise and sensitivity to strain. In the 30-layered magnetite print, the sensitivity is the largest; however, by using the same inductor coil, the level of signal-to-noise ratio for 30 layers is larger compared to other layers. For example, the error/noise in a 30-layered print is around ±4.2 µH, whereas the error/noise in a 20-layered print is around ±2.2 µH. This could be due to the excess porosity and voids present in the print, which restricts magnetic moment orientation. In the 20-layered square design, the sensitivity is greatest beyond 0.2 µε and saturates at 0.4 µε. The level of error in the 20-layered print is the best compared to other layered designs. The 10-layered square design shows the least sensitivity to strain and the lowest measured magnetic field. This is expected, as there is less magnetite material on the surface; hence, it will have a lower field and reduced sensitivity. The level of error is quite large, which is surprising, as it would contain less porosity than the other designs, such as the 20-layered magnetite. There is a difference in maximum inductance saturation as more material is layered on top. For the 10 and 20 layers, the inductance is saturated at around 0.2 µε, whereas 30-layered magnetite has an inductance maximum at around 0.225 µε. This is due to the amount of material on top, as more strain is needed to orientate the magnetic moments to reach maximum strain when the coil is placed on top. Coating a thin layer of PDMS or acrylic polymer on the 30-layered magnetite could help with reaching the maximum inductance.

#### 3.4.2. Bending Test

For structural health monitoring of CFRP, the inkjet-printed sensor will be directly printed onto the CFRP rather than paper. Therefore, a magnetite coil design was directly printed on top (Figure 14). The change in inductance as a function of applied strain was measured, as previously performed for the sensors on paper, but this time, it was found that smaller bend radii were required to achieve a higher strain.

The inductance measurements were carried out using a magnetite coil sensor, printed on CFRP using a 200 coil turns inductor and a 10 µF capacitor in series. While the results in Figure 15 show that there is a change in inductance as a function of radii, the initial measurement (baseline) showed that there was a drift over time; therefore, an average was taken. The change in inductance is higher compared to the magnetite coil sensor on a photo paper substrate. In comparison, ref. [9] have reported a resolution of 0.211 μStrain with CoFeB ribbon, while this work shows that the coil has a resolution of 0.12 μStrain. While different types of inductors were used, such as the number of turns, it does show that the sensor used here is much lower compared to literature due to a higher magnetostriction of CoFeB than magnetite.

## 4. Conclusions

Magnetic sensors were made from 20% magnetite NP and DMP solvent, and the designs were made and selected to print using the JetLab 4 inkjet printing system. These included a coil, uniaxial patch, grid and rectangular block to test the inductance sensitivity as a function of strain.

The print was analysed by optical microscope, SEM and EDS. Print analysis shows that the print had a resolution of 132 DPI. This was lower than in literature studies, where they achieved a resolution of 180 DPI. This is because the jetting parameters were set to print larger droplets to ensure magnetic NPs were jetted and not to let the NPs sink to the bottom of the reservoir. EDS shows magnetite and nickel composition, where iron and oxygen were present in magnetite droplets, and nickel was present in nickel droplets.

The coil design was selected, and the best ink for strain bending was magnetite. The coil design far exceeded other designs that were thought to have anisotropy due to the print design such as in uniaxial patch design. This may be due to the print pattern where the coil design contained larger gaps between the coil turns; therefore, it may improve the field through increased demagnetisation. It was evident that the greater number of layers of magnetite increased inductance as a function of strain. For example, the 20-layered square patch had a greater response to strain than the 10-layered square magnetite patch. However, beyond 20 layers, the print quality was reduced even though the magneto-mechanical response was slightly higher. This increased the level of noise that would not be suitable for SHM applications.

By continuing from inkjet printing on photo paper, strain by bending over a known curvature was performed on a CFRP sample with a magnetite coil printed directly on top. The results showed that printing directly on the surface of CFRP, performed better than on paper substrate. This could be due to the paper substrate absorption of the magnetite and higher wetting angle on kapton than CFRP. Acrylic spray or PDMS coating can be used to protect the magnetite print on CFRP and prevent damage to the environment.

As magnetostrictive materials are known to provide SHM of CFRP, this paper has contributed to developing a sensor via inkjet printing magnetostrictive material and shown how changing the design and number of layers affects the change in inductance under applied strain. This work provides scope for future research and development into SHM and printing magnetic materials.

## Figures and Tables

**Figure 1 sensors-24-04657-f001:**
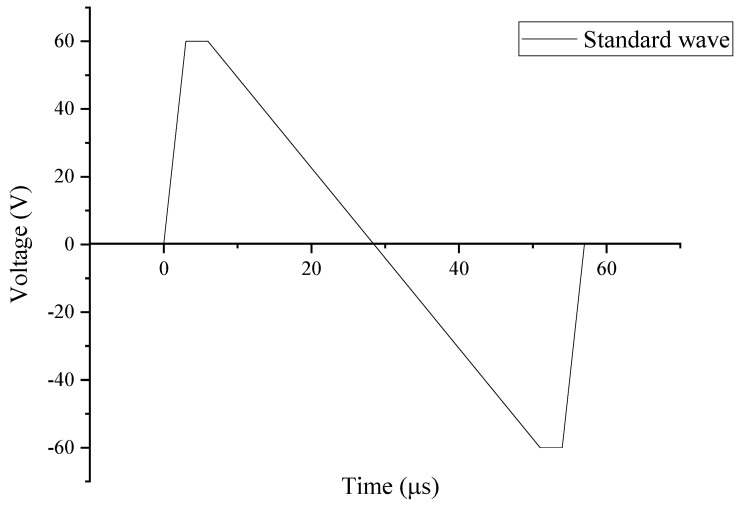
JetLab signal input standard wave for each droplet.

**Figure 2 sensors-24-04657-f002:**
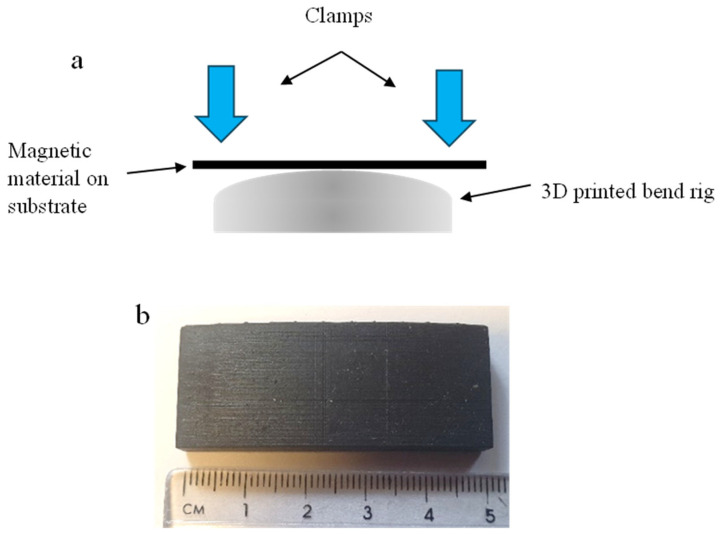
(**a**) Bending test on a known radius of curvature (**b**) 3D printed bend rig.

**Figure 3 sensors-24-04657-f003:**
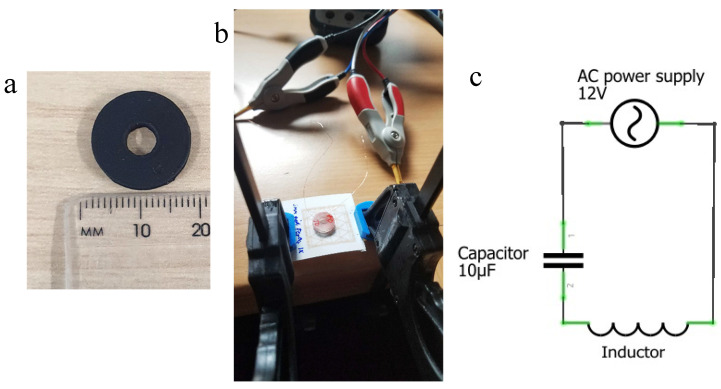
(**a**) Dimension of the air core coil holder used for inductance measurements, (**b**) inductance measurement with coil and clamp on 3d printed bend rig and (**c**) circuit schematic with inductor, capacitor (where 1 and 2 are positive and negative connection) and AC power supply in series.

**Figure 4 sensors-24-04657-f004:**
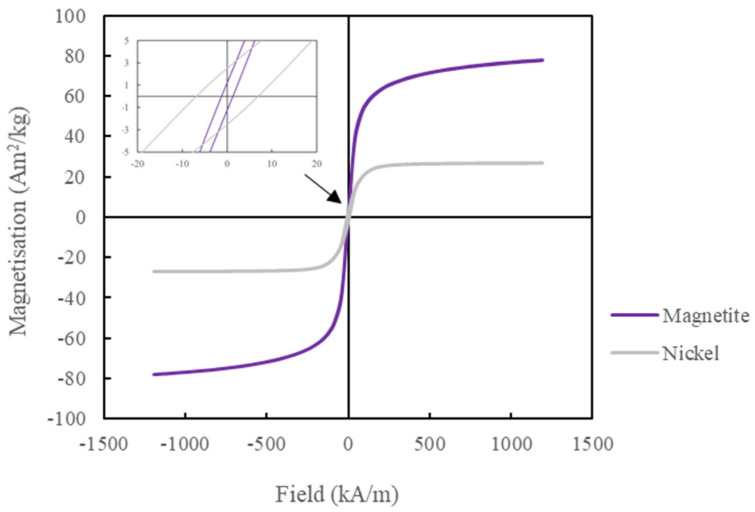
Hysteresis loop of magnetite and nickel NP from −1200 to 1200 kA/m field.

**Figure 5 sensors-24-04657-f005:**
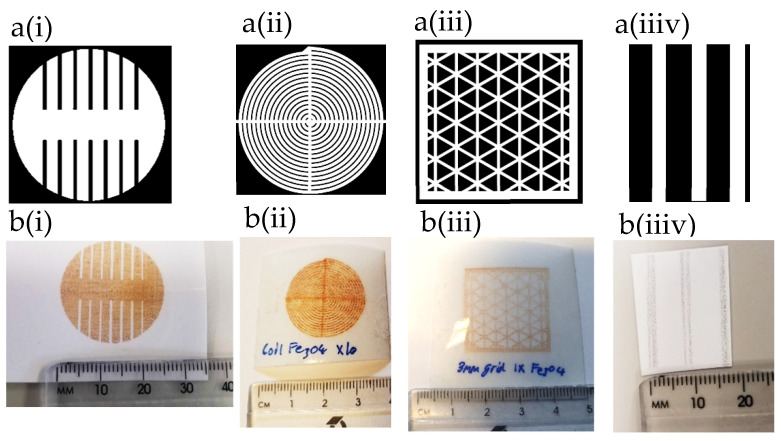
(**a**) image and (**b**) printed design: (**i**) magnetite uniaxial patch, (**ii**) magnetite coil and (**iii**) grid design and (**iiiv**) nickel lines print.

**Figure 6 sensors-24-04657-f006:**
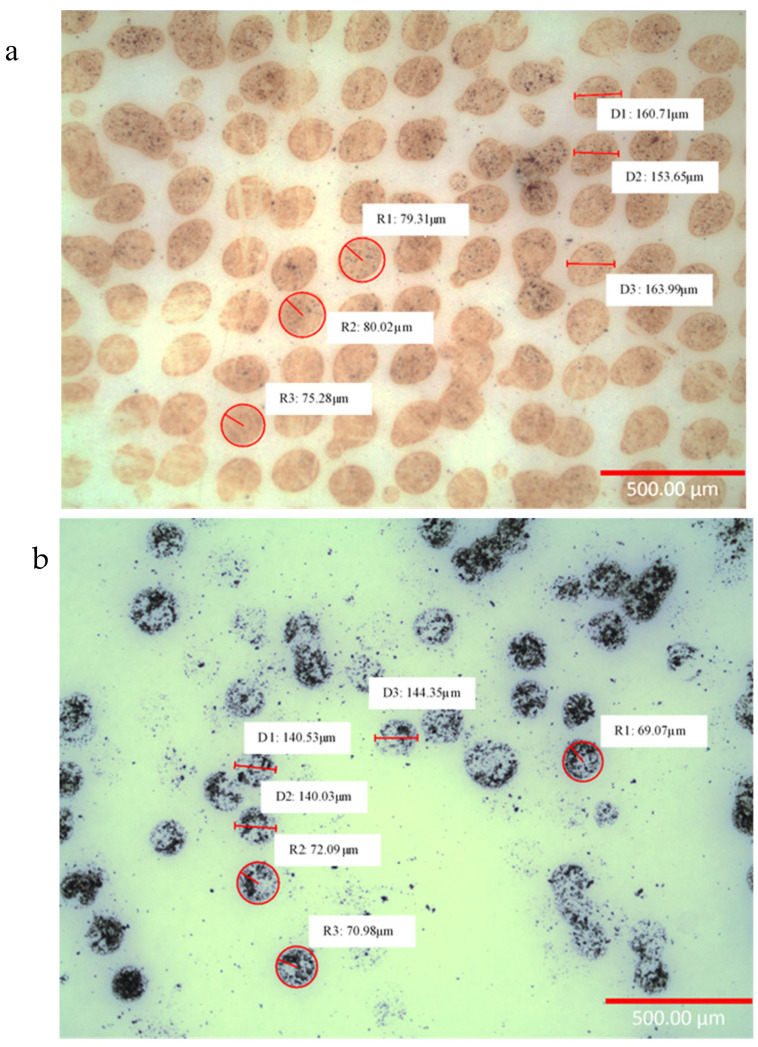
Optical microscope of (**a**) magnetite uniaxial patch design on paper and (**b**) nickel coil design on paper.

**Figure 7 sensors-24-04657-f007:**
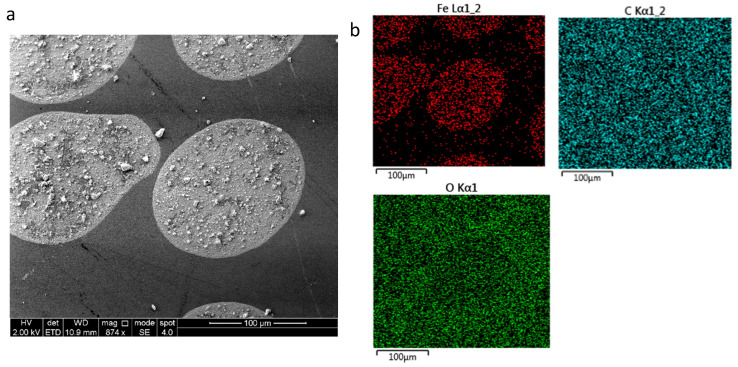
(**a**) SEM image of magnetite on paper at 2 kV, (**b**) EDS element map at 10 kV, (**c**) SEM spectrum label at 2 kV, (**d**) EDS spectrum 2 at 10 kV and (**e**) EDS spectrum 3 at 10 kV.

**Figure 8 sensors-24-04657-f008:**
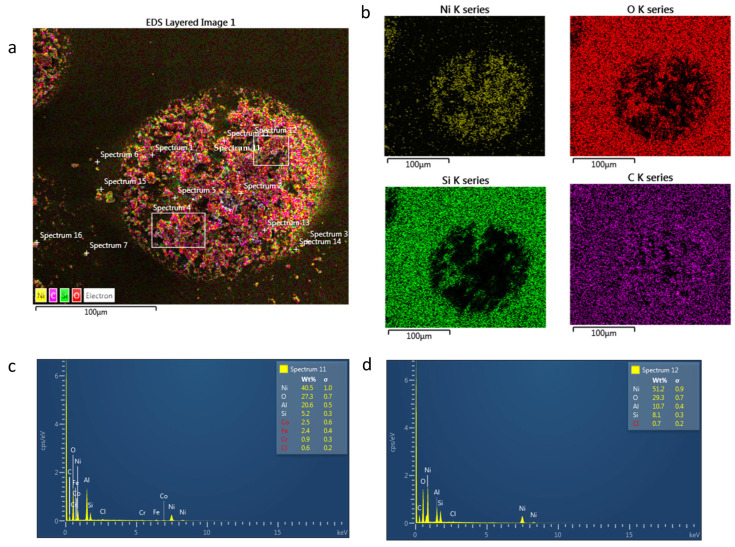
(**a**) Nickel droplet on paper SEM at 2 kV, (**b**) EDS layered map at 10 kV, (**c**) nickel EDS spectrum 11 and (**d**) nickel EDS spectrum 12 at 10 kV.

**Figure 9 sensors-24-04657-f009:**
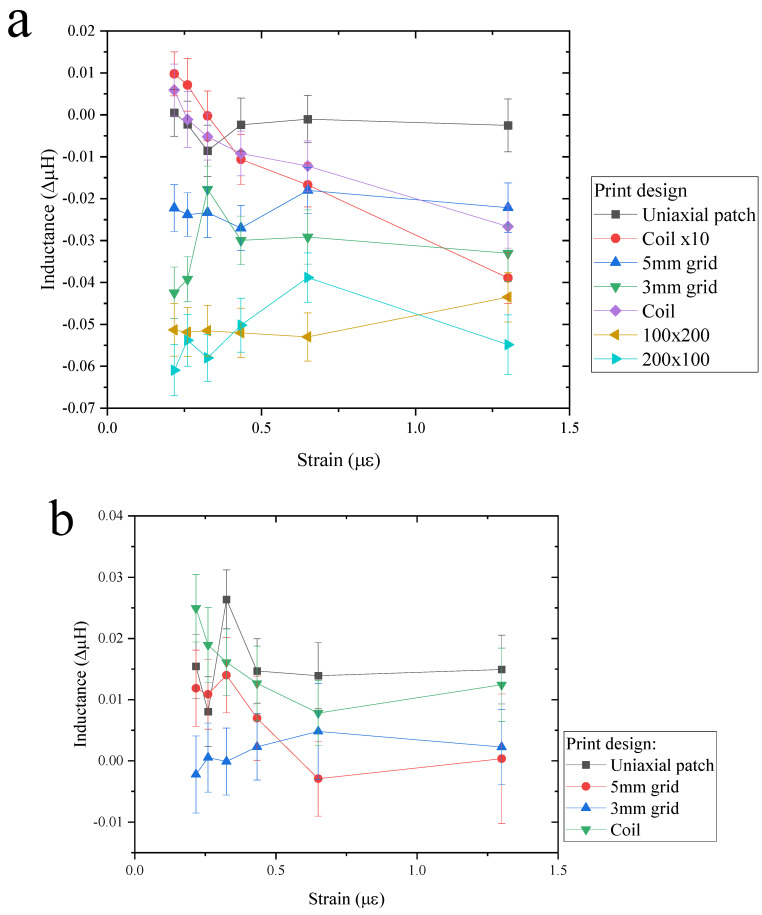
Change in inductance as a function of strain from 0.22 µε to 1.3 µε for (**a**) magnetite print designs and (**b**) nickel print designs.

**Figure 10 sensors-24-04657-f010:**
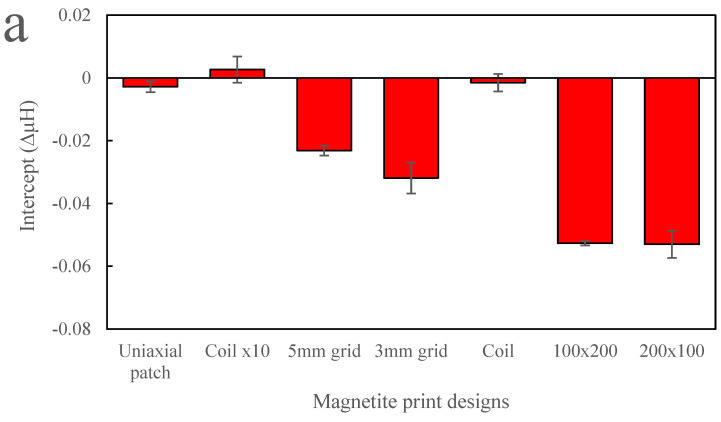
Magnetite print designs linear fitting showing (**a**) intercept, (**b**) gradient and (**c**) R-squared value.

**Figure 11 sensors-24-04657-f011:**
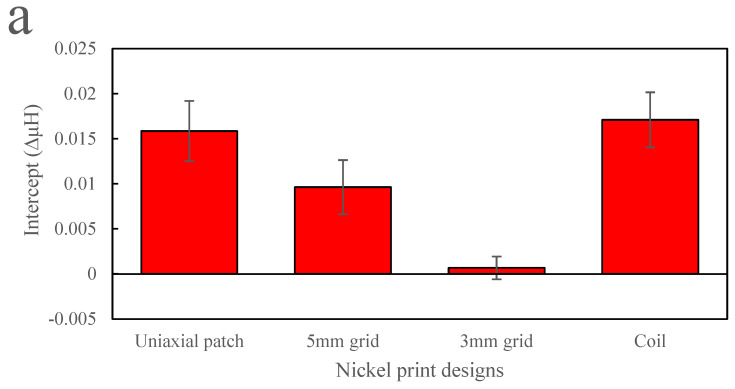
Nickel print designs linear fitting graph showing (**a**) intercept, (**b**) gradient and (**c**) R-squared value.

**Figure 12 sensors-24-04657-f012:**
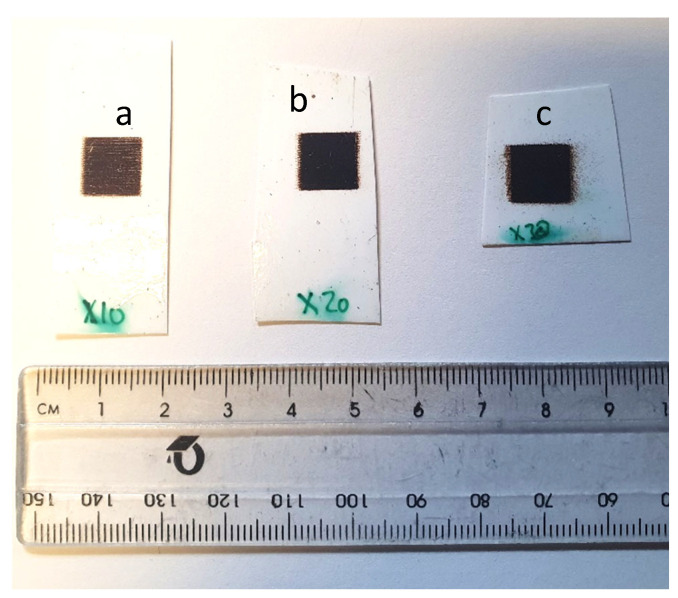
Magnetite print in (**a**) 10, (**b**) 20, (**c**) 30 layered square design.

**Figure 13 sensors-24-04657-f013:**
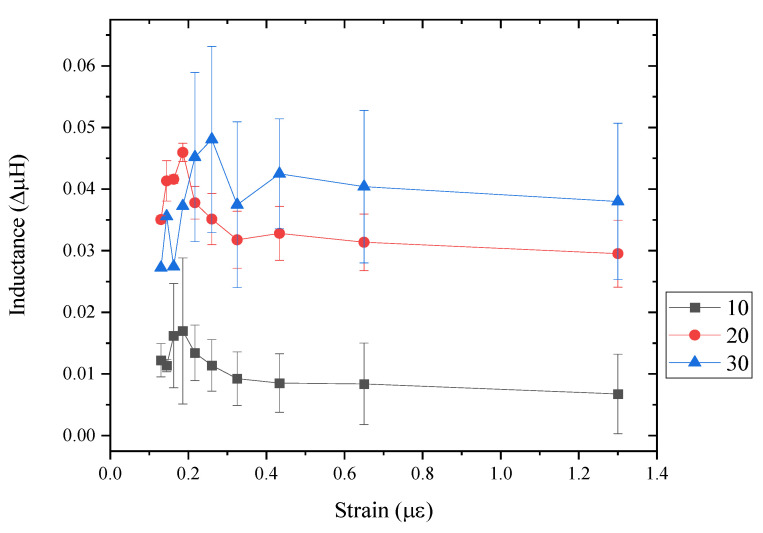
Bending test of magnetite square designs of 10, 20 and 30 layers measuring the inductance as a function of strain.

**Figure 14 sensors-24-04657-f014:**
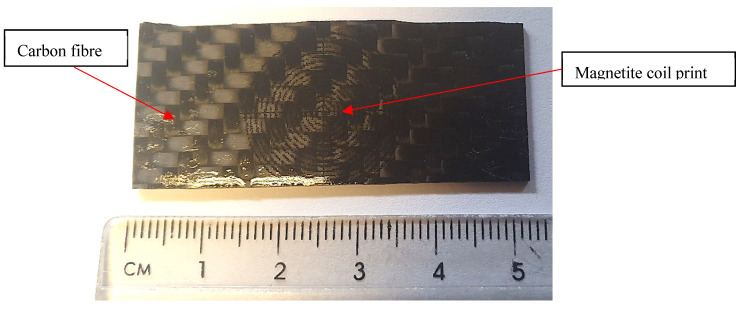
Magnetite coil print on CFRP.

**Figure 15 sensors-24-04657-f015:**
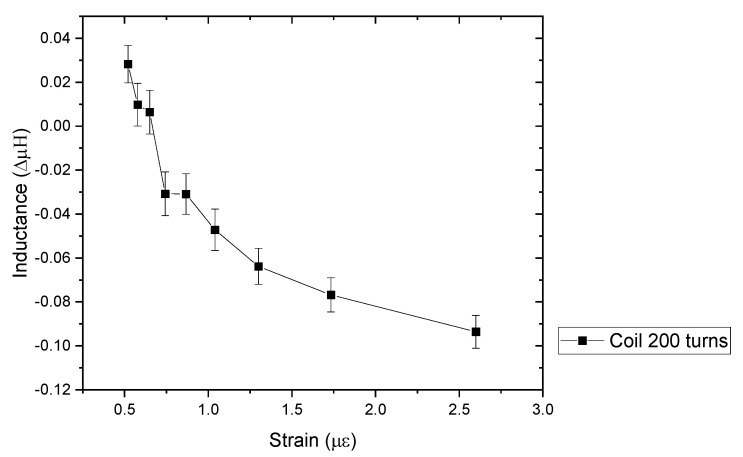
Inductance of coil print on CFRP with 200 coil inductor.

**Table 1 sensors-24-04657-t001:** Nickel and magnetite ink properties.

Metal NP	Solvent Composition	Viscosity at 298 K (cP)	Density (g/cm^3^)	Surface Tension (mN/m)	Particle Size (nm)	Solvent Evaporation Temperature (K)
Ni (2 wt. %)	NMP and water	7.5–10	2.07	72.8 (water)	80–100	473 (NMP)
Fe_3_O_4_ (20 wt. %)	Organic Solvent (DMF), IPA Ethanol, Water (ddH_2_O)	7.5–10	5.17	72.8 (water)	50–80	426 (DMF)

**Table 2 sensors-24-04657-t002:** Magnetite and nickel saturation magnetisation, remanence and coercivity calculated from the hysteresis loop.

Material	M_s_ (Am^2^/kg)	M_r_ (Am^2^/kg)	H_c_ (kA/m)
Magnetite	76.5	1.25 ± 0.02	1.22 ± 0.02
Nickel	27.1	2.53 ± 0.01	6.56 ± 0.10

**Table 3 sensors-24-04657-t003:** Viscosity measurement for magnetite and nickel NP.

Metal NP	Solvent	Temperature (K)	Viscosity (cP)
Magnetite	DMF, IPA and water	294	1.92
Nickel	NMP and water	293.6	2.06

## Data Availability

Unavailable due to privacy.

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
