# Peer review of "Inkjet Printing Magnetostrictive Materials for Structural Health Monitoring of Carbon Fibre-Reinforced Polymer Composite"

_sensors, 2024, doi:10.3390/s24144657_

Round 1

Reviewer 1 Report

Comments and Suggestions for Authors

Structural defects in iron products are traditionally examined using magnetic flaw detection. Modern technologies for applying magnetic ink prints allow such studies in carbon fiber products. In particular, their deformation can be controlled. In this paper, the authors describe their experience in the development and use of magnetic ink based on magnetite and nickel nanoparticles. The authors test paints for viscosity and solvent evaporation, as well as their behavior on various surfaces. According to a number of parameters, magnetite nanoparticles turned out to be more attractive for magnetic ink. Several printed patterns have been studied and the optimal print having the best field response to deformation has been established. For this purpose, tests were carried out on the inductance of various imprints on carbon fiber plastic depending on the magnitude of bending deformation. This material is interesting and fully corresponds to the profile of the journal. However, before acceptance, the authors need to address a number of shortcomings listed below.

1. Abstract “they have 181.48% higher saturation magnetization and 437.7% lower coercive field than nickel.” Where does such crazy accuracy come from? In Table 2, Ms and Hc are given with an accuracy of 2 and 3 significant digits. Get it in order.

2. Equation 5 is not readable. This makes it difficult to evaluate subsequent results regarding inductance.

3. Text information in Fig. 7, 8, 10, 11 are of poor quality or completely unreadable. This should be fixed.

4. Discussion for Fig. 13 lines 500-502 “From all three-square prints, the change in inductance increased from an applied strain of 0.13 με to 0.2 με, then the change in inductance reduced as applied strain increased from 0.25 με. This pattern was present in all print layers, which suggest that the print is sensitive to strain even at 0.13 με." These statements are doubtful, because error indicated in Fig. 13 exceeds the indicated changes. Those in view of Fig. 13, the statement “deformation does not affect the inductance at all” also looks reliable.

5. The phrase in the conclusion “A selection of magnetic sensor designs was selected **” looks clumsy.

6. Be careful in notation. For example, line 61 “d33” should be subscripted. Check this throughout the text.

Author Response

Comments 1: Abstract “they have 181.48% higher saturation magnetization and 437.7% lower coercive field than nickel.” Where does such crazy accuracy come from? In Table 2, Ms and Hc are given with an accuracy of 2 and 3 significant digits. Get it in order.

Response 1: The percentages have been corrected and replaced in text.  The data have been replaced with 3 significant figures 

Comment 2: Equation 5 is not readable. This makes it difficult to evaluate subsequent results regarding inductance.

Response 2: The equation has been made larger and increased spacing 

Comment 3: Text information in Fig. 7, 8, 10, 11 are of poor quality or completely unreadable. This should be fixed

Response 3: Text information have been updated in Fig. 7, 8, 10 and 11

Comment 4: Discussion for Fig. 13 lines 500-502 “From all three-square prints, the change in inductance increased from an applied strain of 0.13 με to 0.2 με, then the change in inductance reduced as applied strain increased from 0.25 με. This pattern was present in all print layers, which suggest that the print is sensitive to strain even at 0.13 με." These statements are doubtful, because error indicated in Fig. 13 exceeds the indicated changes. Those in view of Fig. 13, the statement “deformation does not affect the inductance at all” also looks reliable.

Response 4: Updated the discussion and mentioned the error is larger than the change in inductance for 10 and 30 layered prints 

Comment 5: The phrase in the conclusion “A selection of magnetic sensor designs was selected **” looks clumsy.

Response 5: Updated conclusion and mentioned each designs as sensor in the manuscript 

Comment 6: Be careful in notation. For example, line 61 “d33” should be subscripted. Check this throughout the text.

Response 6: Changed notation on line 61 and throughout the paper 

Reviewer 2 Report

Comments and Suggestions for Authors

I enjoyed reading the paper: a fine experimental description. Not something earth-breaking, but a complete work for an important issue. I suggest publication.

Comments on the Quality of English Language

It seems that the manuscript has been written or corrected by a British colleague

Author Response

Thank you for your kind comments to my paper. Glad you enjoyed reading without any additional issues.  

Reviewer 3 Report

Comments and Suggestions for Authors

The authors present a work focused on the printability of two magnetic inks. They evaluated some sensing designs based on magnetic alterations due to mechanical deformations. The research topic is interesting but there are some major issues that prevent its publication:

1. Although there is enough workload for a paper in the journal, the authors do not provide clear objectives and motivation for the work.

2. The contributions of the work to the state of the art are not clear. There is no connection between literature review, gaps of knowledge and what is done here.

3. Some of the figures are of a really bad quality.

Comments on the Quality of English Language

The writing must be revisited as there are several typos throughout the document.

Author Response

Comment 1: Although there is enough workload for a paper in the journal, the authors do not provide clear objectives and motivation for the work.

Response 1: The main aims and objectives have been updated in the introduction chapter

Comment 2: The contributions of the work to the state of the art are not clear. There is no connection between literature review, gaps of knowledge and what is done here.

Response 2: Updated SHM literature review and identified gaps in knowledge. Comparison between literature and this work have been updated in text 

Comment 3: Some of the figures are of a really bad quality.

Response 3: Figures have been edited by increasing the brightness and contrast

Round 2

Reviewer 1 Report

Comments and Suggestions for Authors

The authors have worked sufficiently in accordance with my comments. The article can now be accepted.

Reviewer 3 Report

Comments and Suggestions for Authors

The authors addressed my previous concerns.

Comments on the Quality of English Language

Could be improved.